# EFloat: Entropy-coded Floating Point Format for Deep Learning

## Abstract

In a large class of deep learning models, specifically vector embedding models in NLP, we observe that floating point exponent values tend to cluster around few unique values, presenting entropy encoding opportunities. The proposed EFloat floating point number format encodes frequent exponent values and signs with Huffman codes to minimize the average exponent field width while keeping the original exponent range unchanged. Saved bits then become available to the significand increasing the EFloat numeric precision on average by 4.3 bits compared to other low-precision floating point formats of equal bit budget. The EFloat format makes 8-bit and smaller floats practical by preserving the full exponent range of a 32-bit floating point representation. We currently use the EFloat format for compressing and saving memory used in large NLP deep learning models while I/O and memory bandwidth savings in GPUs and AI accelerators are also possible. Using RMS-error as a precision metric, we demonstrate that EFloat provides more accurate floating point representation than other formats with the same bit budget. EF12 with 12-bit budget has less end-to-end application error than the 16-bit BFloat16. EF16 RMS-error is 17 to 35 times less than BF16 RMS-error for a range of datasets. Using the NDCG metric for evaluating ranked results of similarity and dissimilarity queries in NLP, we demonstrate that EFloat matches the result quality of other floating point representations with larger bit budgets.

## 1 Introduction

As natural language processing (NLP) models expand their capabilities, complexity, and training costs, the model sizes have been increasing dramatically. For example, state-of-the-art transformer-based NLP models such as BERT (Vaswani et al. (2017)), Megatron-LM (Shoeybi et al. (2020)), Open AI GPT-3 (Brown et al. (2020)), or Google Switch-C Transformers (Fedus et al. (2021)), contain from hundreds of millions, to even trillion parameters (Hoefler (2020); Fedus et al. (2021)). Although NLP model compression is a very active area of research (Section 6), its current focus is on model inference scenarios, in which reduced precision and integer quantization are commonly used, given that the original model need not be restored.

The primary goal for this work is to explore compression strategies for large vector embedding models such that one can recover or minimize the loss in the original model, or use the same compressed model in the inference phase. The *database embedding (db2Vec)*, a vector embedding technique designed to develop semantic models from multi-modal relational database tables (Bordawekar and Shmueli (2016, 2017)), forms the impetus behind this exploration. Db2Vec differs from its NLP counterparts, such as Word2Vec (Mikolov et al. (2013)) and GloVe (Pennington et al. (2014b)), in that its source data follows the relational data model (Date (1982)) (the source data is not a natural language document but a relational database table). Considering the relational database tables can

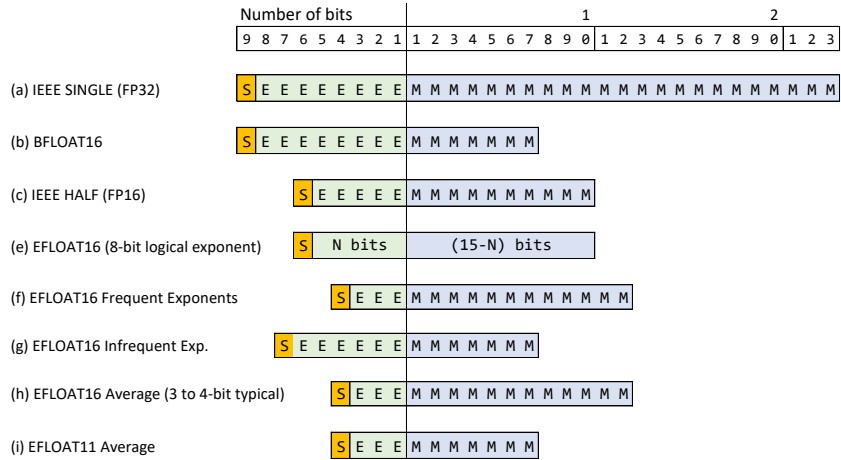

Figure 1: Floating point formats are compared. EFloat has a fixed total width, but the boundary between the exponent and the significand is variable (e). The exponent is entropy coded, providing an average of 4.3 extra bits of precision to the significand (e.g., (h)), while keeping the logical exponent range at 8 bits, same as that of FP32. EFloat has greater precision and range than the existing FP formats having the same bit budget.

be very large (e.g., billions of rows in a table) with a large number of unique tokens, it leads to a much larger vocabulary than a traditional natural language document. As a result, trained db2Vec models can be very large. Any trained vector embedding model is a snapshot of it's weight matrices and consists of weight values represented typically with IEEE 32-bit single-precision floating point (**FP32**) format. Therefore, we focus on compression approaches that exploit different low-precision floating point formats.

Existing low-precision floating-point(FP) formats make a tradeoff between the number of exponent and significand bits. An FP number is of the form

$$-1^{signbit} \times 2^{exponent-bias} \times significand$$

The exponent largely determines the range of minimum and maximum values representable by the format and the significand width determines the precision (A constant bias is added to exponents to make them all positive integers which simplifies magnitude comparisons.) For example, the BFloat16 (**BF16**) format with an 8-bit exponent and 7-bit significand has a wide range but low precision when compared to FP32 (Wang and Kumar (2019)) and IEEE 754-2019 Half-precision (**FP16**) as illustrated in Figures 1(a,b,c). On the other hand, FP16 with a 5-bit exponent and 10-bit significand has a greater precision but a tighter range than BF16 (IEEE (2019)).

In this work, we introduce a new low-precision FP format, *EFloat (EFn)*, that uses entropy-coded variable-width exponent and a variable-width significand with a total FP bit budget $n$, as illustrated in Fig.1(e). Our design is motivated by a key pattern that we observed across a wide range of vector embedding models: post-training, these models use only few of the $2^8 = 256$ unique exponents available in FP32 and with a bell-shaped distribution caused by a certain class of non-linear activation functions used in model training. The EFloat design exploits this behavior and assigns the least number of exponent bits to most common exponent values, without losing the exponent range of the original floating point value.

The proposed EFloat format has the following benefits:

- Reduced-bit representation of *any* floating point format (e.g., FP32, FP16), by using fewer exponent bits to map the **same** exponent range as the original value.

- For a given bit budget (e.g., 16), EFloat provides more accurate representation of the FP32 values than BF16 and FP16 by using fewer exponent bits to capture the same range as before, and then using the remaining bits to increase significand precision.

- The format is suitable for both memory and bandwidth **compression** and reduced-bit **computations** over *pre-trained* vector embedding models. Software implementation trades

compute cycles with capacity and I/O bandwidth savings. A factor of 3 reduction in memory footprint is achieved converting FP32 values to EF11. Hardware conversion is possible for FPn to EFn and vice versa with simple Static RAM based lookup tables.

- For a given dataset, many different FP to EF conversion tables are possible. Tables may be optimized for maximum significand width (highest exponent compression) at the expense of worse precision for few floats with infrequent exponents (less significand bits for outliers) and vice versa.

- Since vector embedding models are used in a wide array of NLP transformer architectures, the EFloat format can be used for a much wider (and more space consuming) class of NLP models.

In Section 2, we first present the analysis of various vector embedding models. The EFloat format is presented in Section 3. Section 4 describes key steps in conversion between EFloat and other FP formats. Section 5 presents an error analysis of various EFloat widths (EFn) against BF16 and FP16. In Section 6, a review of related work on model compression and floating-point formats for deep learning is presented. Finally, Section 7 presents conclusions and outlines future directions.

## 2 Analyzing vector embedding models

Vector embedding models are extensively used in natural language processing (NLP) to capture and exploit semantic relationships of word entities (e.g., words, sentences, phrases, paragraphs, or documents). A trained vector embedding model consists of a set of vectors, each vector encoding a *distributed* representation of inferred semantics of a word entity, i.e., a single vector captures different attributes of the inferred semantics (Hinton et al. (1986)), created, in part, by contributions by other word entities. Every vector embedding model implements some variant of the *log-bilinear* language (LBL) model that predicts the probability of the next word $w_i$ given the previous words (*context*) (Hinton (2013); Almeida and Xexeo (2019); Bender and Koller (2020)). The LBL model first predicts a real-valued vector representation of a word by *linearly* combining the real-valued vector representations of its context words. Then the distributed representation of the word is computed based on the similarity between the predicted representation and the representations of all words in the vocabulary. This step is accomplished using the *normalized exponential* or *Softmax* function over the associated feature vectors. The output of the Softmax function is the probability distribution over $V$ different possible outcomes, where $V$ is the vocabulary size.

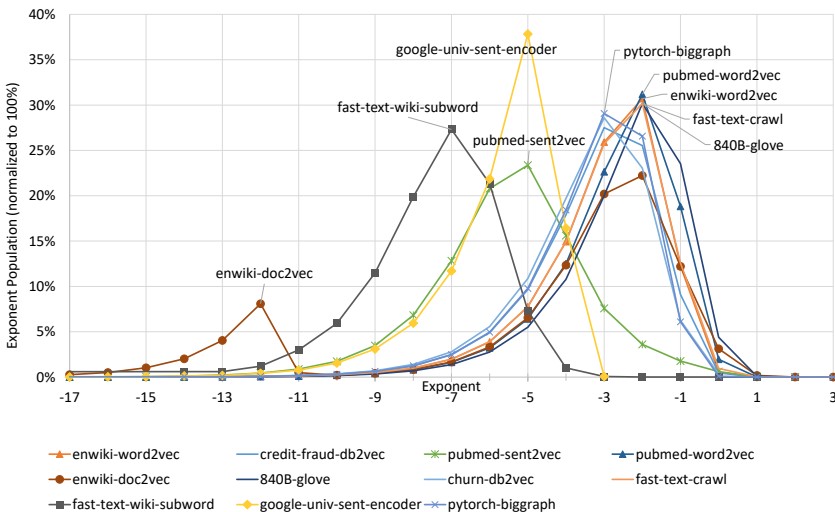

Figure 2: Histogram of the exponent fields of 32-bit floating-point (FP32) values found in vector-embedding and related NLP models. Only the db2Vec, word2vec, doc2vec, and sentence-encoder models were generated. Others were downloaded as publically available pretrained models.

Figure 2 presents histograms of exponent values in multiple pre-trained vector embedding models, where the X-axis represents exponent values (from the 8-bit exponent portion of a 32-bit IEEE 754

floating point value). For each X-axis value, the Y-axis represents normalized number of occurrences of that exponent value, i.e., a histogram. Vector embedding and related NLP models presented in Fig. 2 include word embedding (word2Vec) (Mikolov et al. (2013); Zhang et al. (2019)), sentence (sent2Vec) and document embedding (doc2Vec) (Le and Mikolov (2014); Chen et al. (2019)), GloVe (Pennington et al. (2014a)), subword embedding (FastText) (Bojanowski et al. (2017a,b)), database embedding (db2Vec), graph embedding (PyTorch BigGraph Lerer et al. (2019)), and Google's transformer-based universal sentence encoder (Cer et al. (2018); Google (2021)) using the Brown corpus ( Brown-corpus (2021)). All these models implement different variations of the LBL model. The word2Vec based models, e.g., word2Vec, sent2Vec, doc2Vec, db2Vec, and FastText, use a neural network with different versions of Softmax as the activation function. GloVe, on the other hand, is a count-based optimization approach that uses a word co-occurrence matrix and weighted least-square as the optimization function. The FastText subword model (Joulin et al. (2016); Bojanowski et al. (2017a)) assigns a vector for every character *n*-gram, using an extended skip-gram model (Mikolov et al. (2013)) and then, words are represented as the sum of these representations. The universal sentence encoder generates embedding vectors for sentences using a standard Transformer architecture that takes word embedding vectors as input and uses a Softmax function to compute attention (Vaswani et al. (2017)). Irrespective of the model type, we observe that exponent values cluster around a certain range of values, and display a distinct *peak*. The only exception is the doc2Vec model that exhibits two peaks as the doc2Vec first builds fine-grained embeddings for words and then uses them to build embeddings for coarser-grained entities such as paragraphs via concatenating and averaging individual word vectors which results in a smaller second peak as observed in Fig. 2.

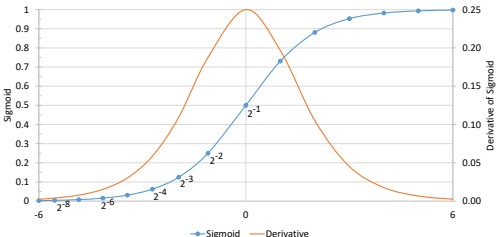
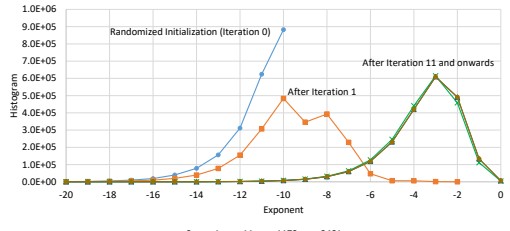

Figure 3: The Sigmoid $\sigma(x)$ curve and its gradient. The floating-point (FP32) exponent of few neural weights are overlaid on $\sigma(x)$.

Figure 4: Exponent population (FP32) of model weights during the training of churn-db2vec model, for iterations 0 to through 2481

The Softmax family of activations functions used in vector embedding models is responsible for the clustering behavior of exponents (Figure 2). To understand the reasons, let us delve deeper into the training of an embedding model. For illustration purposes, we use database embedding (db2Vec) of the Telcom Churn data (IBM (2020)) as an example. db2Vec is an adaptation of the word2Vec approach, and has been designed to build an embedding model from structured database tables that adhere to the relational data model. Like word2Vec, db2Vec also uses Skipgram with Negative Sampling (SGNS) as the training approach. The SGNS approach uses a binary classifier based on the logistic (Sigmoid) function instead of using the Softmax-based predictor. The overall training process involves multiple back-propagation iterations to update model weights using the gradient of the Sigmoid function. Weights get updated iteratively during the back-propagation process by the error computed for that iteration. Practically, the error is computed using the gradient of the activation function. During model training, we observe that the weights rapidly converge (Fig.4) to their final values. Their exponents are substantially clustered at the slope of the Sigmoid curve, the $2^{-8}$ to $2^0$ output range of Sigmoid, as evidenced by Figures 3 and 4. Training eliminates smaller exponents from the model because the activation function output is practically zero for any input value when weights are small, and large exponents are non-existent of normalization of weights.

# 3   The EFloat format: EF$n$

The key idea behind the EFloat format is the variable-width encoding of exponents using the well-known Huffman algorithm. Frequency of unique exponent values in the dataset determine the coded-exponent widths which may vary between as small as 1-bit and some software configurable maximum, e.g., 8-bit (Figure 1(e,f,g)). Thanks to the entropy coding of the Huffman algorithm,

frequent exponent values are coded with fewer bits and infrequent exponents are coded with more bits as observed in Fig.5.

Bits saved from the exponent become part of the significand, therefore increasing the floating point precision compared to other float formats with the same bit budget. An $N$-bit coded-exponent in an EF16 float results in a (15-$N$)-bit significand as shown in Fig.1(e). Since Efloats with frequent exponents have wide significands, the entire dataset has a greater precision on average. Efloats with infrequent exponents have narrow significands. But, their contribution is relatively small in the common calculations used in model training and inferencing, such as dot-products, vector-sums, and cosine-similarity (EFloat precision is quantified and compared to prior formats in Section 5.).

EFloat on average have greater precision and range than any other fixed-field FP format with the same bit budget. For example, EF16 with a 3-bit coded-exponent has 12-bits of significand compared to the 7-bit significand found in a BF16 (Figures 1(h,b)). EFloat exponent's logical width is *always* 8-bit, the same as for FP32 and BF16, irrespective of EFloat width. Even for extremely narrow floats such as EF8, the logical exponent width can be 8-bit since encoding compresses the exponent field.

The EFloat format compresses special values of IEEE 754, such as signed zeros and infinities losslessly. NaN are semantically compressed losslessly: converting a NaN to and from FP32 to EFn and vice-versa still results in a NaN. Denormal floats may round to zero since least significant bits of significands are truncated during encoding.

# 4    EFloat encoding and decoding

**The Huffman algorithm:** is a popular lossless compression algorithm used in many compression tools and compressed data formats (Salomon (2004)). Data symbols are encoded with variable-length binary codes whose length are determined by the symbol probabilities in the data stream. The algorithm builds a binary tree with each leaf assigned a symbol. Higher probability symbols are closer to the tree root than others. The path from the tree root to the leaf is the binary coding of the symbol. To demonstrate with a trivial example, the letters A, B, C occuring with probabilities of 0.5, 0.25, and 0.25 may be encoded with the bit patterns 0, 10, and 11, respectively. The algorithm yields 1.5-bit/symbol compression efficiency, better than 8-bits/symbol using an ASCII representation or 2-bits/symbol using a simplistic mapping of the 3 letters to 2-bit integers. Huffman coding is optimal when symbol probabilities are negative powers of 2. However, it is an effective compression method even for non-optimal data distributions. Fig.5 shows the Huffman coded exponent widths as a function of exponent frequencies of a word2vec trained model.

Huffman codes have the *prefix* property which states that no code is a prefix of a longer code (due its tree structure.) As a result, the Huffman code not only encodes the original symbol but the code-length as well. We use this property to locate the bit position of the movable boundary between the exponent and the significand fields when decoding EFloats( Fig.1(e)).

**Length-Limiting:** The basic Huffman algorithm, depending on probabilities, may produce extremely wide codes consuming the entire width of EFloats and more. We use the *Length-Limiting* variant of the Huffman algorithm to set a maximum coded-exponent width (Abali et al. (2020)). In Fig.5, the maximum code-width is set to 8-bits resulting in infrequent exponents encoded with that maximum.

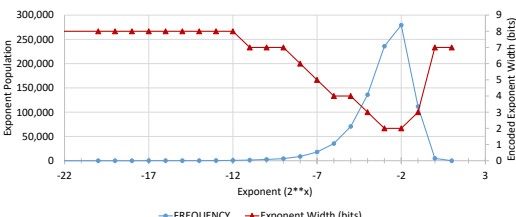

Figure 5: EFloat variable exponent widths are a function of the exponent population (word2vec)

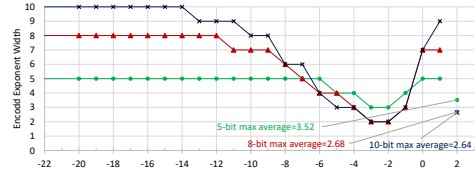

Figure 6: EFloat uses Length-Limited Huffman algorithm to set the maximum width of coded-exponents to 5, 8, or 10-bits

The software-defined limit presents an opportunity to tune the EFloat precision to a particular NLP application requirements. Figure 6 shows the effect of limiting maximum width of coded-exponents to

5, 8, and 10-bits. As the limit is increased, the least frequent exponents are coded with the maximum-width codes, therefore their respective significands lose precision. On the other hand, with increasing limits the most frequent exponents are coded with fewer bits reducing both the minimum and the average coded-exponent widths, therefore their respective significands gain precision. Therefore, EFloat not only compresses the regular floats but for a given EF$n$ budget of $n$-bits the application can optimize the end-to-end precision by adjusting the maximum coded-exponent width.

That the minimum and average code widths are inversely proportional to the maximum code-width may appear counterintuitive (Figure 6). Let the maximum coded-exponent width be $K$-bits ($K = 5$, $K = 8$, etc.). $K$-bits are sufficient to represent $N$ unique exponents in the dataset provided $\lceil log_2(N) \rceil \leq K$ holds. When $K$ is chosen much larger than the minimum $\lceil log_2(N) \rceil$, some codes can have fewer than $K$-bits. Consider a short $L$-bit code such that $L \leq K$ (e.g., assume $K = 8$, $L = 2$ and the short-code = 00.) Due to the prefix property the $L$-bit code consumes $2^{K-L}$ bit patterns out of the $2^K$ maximum possible (e.g., all 8-bit patterns whose 2-bit prefix is 00, 64 patterns in total, are consumed by the code 00.) The remaining $N - 1$ exponents can still be encoded if $N - 1 \leq 2^K - 2^{K-L}$ holds. We observed in practice that increasing the maximum code-width by 2 to 3 bits over the minimum $\lceil log_2(N) \rceil$ gives a good compression ratio.

**EFloat Encoding and the Code-Table:** During the conversion from FP32 to an EF$n$ (e.g., EF16), exponents in the original dataset are histogrammed first, e.g., Fig.2. The histogram representing probabilities of the exponents is the input to the Length-Limiting Huffman algorithm. The output is a 256-entry ($2^8$) code-table indexed by the original 8-bit exponent. Each table entry contains a pair, the variable-width coded-exponent and its width. Note that the code-table is quite small, tens of bytes in practice, since few unique exponents are present in most NLP datasets as Fig.2 shows.

When the sign bits have a skewed distribution, e.g., if they are substantially positive, then the sign bit and the 8-bit exponent may be treated as a single 9-bit integer when histogramming. Thanks to Huffman coding, a skewed sign bit distribution may provide up to one additional bit of precision to the significand.

Using the code-table, the entire dataset is converted from FP32 to the chosen EF$n$ width (e.g., EF16) replacing original exponents with coded-exponents. Least significant bits of the FP32 significand are truncated to match the EF$n$ width. For example, in Fig.5, the algorithm encodes the most frequent exponent with 2-bits. Accounting for the sign bit, this yields a 13-bit significand in EF16 by truncating the bottom 10-bit of the 23-bit significand of FP32. We use the *round-to-nearest* method to provide on average 0.5 bits of additional precision: when the leading bit of the truncated part is 1 the upper part of the significand is incremented +1 provided it doesn't overflow in to the exponent field.

For large datasets, a statistically representative subset may also be used to reduce histogram collection time. When the histogram is known in advance, a pre-built code-table may be used. Pre-built code-tables eliminate the overhead of executing the Huffman algorithm. During training exponents rapidly converge to their final values as observed in Fig.4. The exponent distribution is practically identical for all iterations 11 to 2481, Therefore, a single pre-built code-table optimized for final iterations may serve for all iterations start to finish. The same pre-built table, although suboptimal for early iterations, may be used because significand precision is not as important at that point in time; model weight updates are dominated by exponent updates. Once exponents settled to their final values the significand precision becomes important since model weights updates progressively get smaller.

**EFloat Decoding:** For EFloat to FP32 conversion (i.e., decoding) we use a inverse mapping of the code-table described earlier. A decoder-table indexed by the coded-exponent may be used to decode the original exponent value and the significand's leading bit position in constant time. Each table entry contains the original exponent and width of the coded-exponent. To index the decoder-table with variable-width codes many entries are filled with duplicates. For example, the 2-bit coded-exponent 00 is duplicated 64 times in the table at locations **00**000000 through **00**111111 with each location containing the pair (original exponent and code-width= 2). Duplicating entries is equivalent to having *logical don't care* bits in the index which is a useful in hardware based decoder implementations.

The second element of each table entry contains the EFloat significand width. Since the significand was truncated earlier during the FP32 to EFloat conversion, the missing least significant bits must be padded with zeros to match the original FP32 width.

## 5 Evaluating the EFloat representation

In this section, we evaluate the efficacy of the EFloat format using two sets of experiments. The first set measures the loss of precision in representing FP32 data in BF16, FP16, and EFloat formats with bit budgets from 16 to 8 bits. The second set of experiments compares the quality of ranked results for *similarity* and *dissimilarity* queries using the *Normalized Discounted Cumulative Gain (NDCG)* score for BF16, FP16, and various EFloat formulations. Table 1 presents the list of models used in these experiments, along with their characteristics: model types, model size (stored using FP32), number of unique exponents, range of exponent bits generated by the Huffman algorithm, the average count of exponent bits, and minimum and maximum average count of significand bits. For EF16, the average significand length is 4.3 bits more than BF16 (with 7-bit significand) and 1.2 bits more than FP16 (with 10-bit significand).

Table 1: EFloat characteristics from EF16 to EF8 for different datasets

| Model | Type | Size | Unique exponents | EFn exponent bits Min | Max | Avg. | EFn significand bits (Avg.) Max (EF16) | Min (EF8) |
|---|---|---|---|---|---|---|---|---|
| churn | db2vec | 20 MB | 23 | 3 | 5 | 3.6 | 11.4 | 3.4 |
| crawl | fast-text | 4.3 GB | 30 | 3 | 5 | 3.4 | 11.6 | 3.6 |
| enwiki | word2vec | 9.6 GB | 27 | 4 | 5 | 4.2 | 10.8 | 4.8 |
| MDM | db2vec | 14 GB | 24 | 3 | 6 | 3.6 | 11.4 | 3.4 |
| 840B | GloVe | 5.3 GB | 35 | 3 | 6 | 3.5 | 11.5 | 3.5 |
| wiki-sw | fast-text | 2.2 GB | 22 | 3 | 5 | 3.6 | 10.5 | 3.4 |
| virginia | db2Vec | 222 MB | 24 | 3 | 5 | 3.7 | 11.3 | 3.4 |

The first set of experiments compares the loss of precision due to the least significant significand bits being truncated during conversion from FP32 to various lower-precision formats. Given a low-precision format (e.g., EF16 or BF16), the values are converted back to FP32, and the arithmetic difference, $f^o - f^c$, of the original FP32 value, $f^o$, and the regenerated FP32 value, $f^c$, is computed. This difference represents the precision loss due to conversion. Root Mean Square Error (RMSE) metric is then used to summarize the loss of precision across a dataset of $N$ floats as:

$$RMSE = \sqrt{\frac{1}{N} \sum_k^N (f_k^o - f_k^c)^2}$$

We then compare the errors of BF16/FP16 and EFn by dividing $RMSE_{BF16/FP16}$ by $RMSE_{EFn}$ in Table 2. Ratios greater than $1.0$ indicate that the EFloat error is less than BF16 or FP16 errors. For EF16, across all models, we observe an average RMSE error ratio of 24.1 for BF16, and 1.5 for FP16. Note that for these experimental results, the datasets were encoded with a minimum of 3-bit and a maximum of 6-bit coded-exponents resulting in an average width in the range of 3.4 to 4.2-bits (Table 1). Accordingly, for EF16, the *minimum* significand width is 10-bit which is 3-bit wider than BF16, and of the same length as FP16. Therefore, EF16 has significantly higher precision against BF16 than FP16. Also, Table 2 shows that EF12 has the same to slightly better RMSE than BF16 since the RMSE ratios are in the 1.0 to 2.2 range. Thus, EF12 uses 25% less bandwidth and memory capacity than BF16 for similar floating-point precision.

Note that the RMSE method amplifies larger errors due to the squaring of differences. EFloat coded floating point values with short significands (i.e., those with infrequent exponents) are disproportionately represented in the RMSE summation. However, the true measure of error for vector embedding models will be the evaluation of ranked results for similarity queries for different floating point formats. Unlike the binning in traditional classification inference tasks, ranked results from similarity queries are far more sensitive to numerical precision. We use the *Normalized Discounted Cumulative Gain (NDCG)* metric (Järvelin and Kekäläinen (2002); Wang et al. (2013)), to evaluate the quality of ranked results for different floating point formats. NDCG is widely used in information retrieval and web search to evaluate the relevance of retrieved documents. NDCG is a normalization of the Discounted Cumulative Gain (DCG) measure. DCG is calculated as a weighted sum of the degree of relevancy of the ranked items, where the weight is a decreasing function of the position of an item. NDCG is computed by normalizing DCG by IDCG, which is the DCG measure for a perceived ideal ranking result. Thus, the NDCG measure always lies within [0.0,1.0].

Table 2: BFloat16 (BF16), IEEE Half (FP16), and EF16–8 precision comparisons using RMSE-with-FP32 ratio. Higher is better.

| Model | EF16 | | EF14 | | EF12 | | EF10 | | EF8 | |
|---|---|---|---|---|---|---|---|---|---|---|
| | BF16 | FP16 | BF16 | FP16 | BF16 | FP16 | BF16 | FP16 | BF16 | FP16 |
| churn | 22.5 | 1.4 | 5.6 | 0.4 | 1.4 | 0.09 | 0.3 | 0.02 | 0.08 | 0.005 |
| crawl | 34.6 | 2.2 | 8.6 | 0.5 | 2.2 | 0.1 | 0.5 | 0.03 | 0.1 | 0.008 |
| enwiki | 16.9 | 1.0 | 4.2 | 0.3 | 1.0 | 0.07 | 0.3 | 0.02 | 0.06 | 0.004 |
| MDM | 27.9 | 1.8 | 6.9 | 0.4 | 1.8 | 0.1 | 0.4 | 0.03 | 0.1 | 0.007 |
| 840B | 25.0 | 1.6 | 6.3 | 0.4 | 1.6 | 0.09 | 0.4 | 0.02 | 0.09 | 0.006 |
| wiki-sw | 22.0 | 1.4 | 5.5 | 0.3 | 1.2 | 0.08 | 0.3 | 0.02 | 0.08 | 0.005 |
| virginia | 19.6 | 1.2 | 4.9 | 0.3 | 1.2 | 0.08 | 0.3 | 0.02 | 0.07 | 0.004 |

For a given vector embedding model, we choose $q = 20$ randomly selected distinct query points. For each query point, we compute similar and dissimilar points by computing cosine similarities over the corresponding vectors. For similarity queries, the result contains a list of points sorted in decreasing order of their similarity scores (most similar pair of items will have score closer to 1.0), and for dissimilarity queries, the result list is sorted in increasing order of their similarity scores (most dissimilar pair of items will have score closer to -1.0). For each query point, we run similarity and dissimilarity queries for different floating point formats, and use the top $k = 10$ results for each test to compute the NDCG score, (**NDCG@10**). In our evaluation, we use the ranked results for FP32 as the baseline for calculating the IDCG. For each model, we report the average NDCG@10 score computed over 20 query points using BF16, FP16, and various EFn from EF16 to EF8.

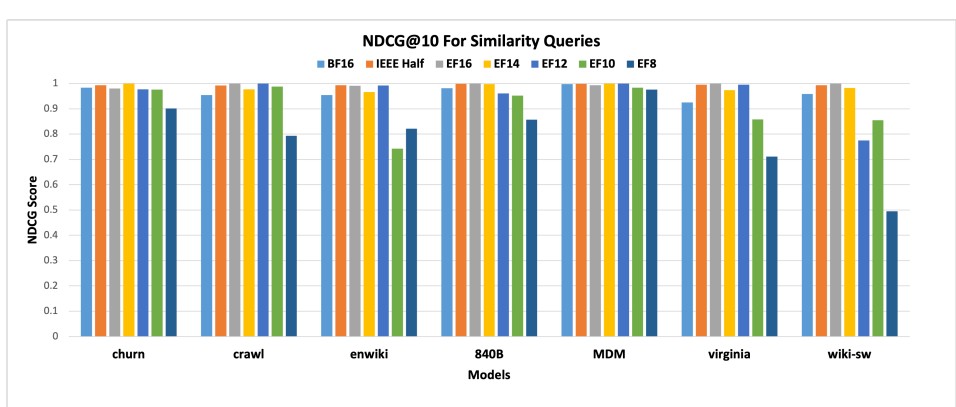

Figure 7: Evaluation of similarity query accuracy using NDCG score across different floating point formats. Higher score (closer to 1.0) is better.

Figure 7 presents NDCG10 results for similarity queries, and Figure 8 presents NDCG10 results for dissimilarity queries. For both similarity and dissimilarity queries, EF16 matches or exceeds the quality of BF16 or FP16 (in particular, among the three formats, BF16 provides the worst qaulity results). Furthermore, EF14 and EF12 provide similar quality results as EF16 in many instances. The two lower-precision EFn, EF10 and EF8, consistently generate the least quality results.

In summary, results from the two sets of experiments (Table 2, and Figures 7 and 8), conclusively demonstrate that: (1) Given a bit budget, EFloat has higher accuracy than other formats, (2) In many scenarios, EFn with reduced bit budget (e.g., EF14 or EF12) provides results of quality comparable to higher precision formats, e.g., BF16, and FP16. These results validate the design of the EFloat format, and demonstrate that EFloats can be used for compressing and computing using vector embedding models.

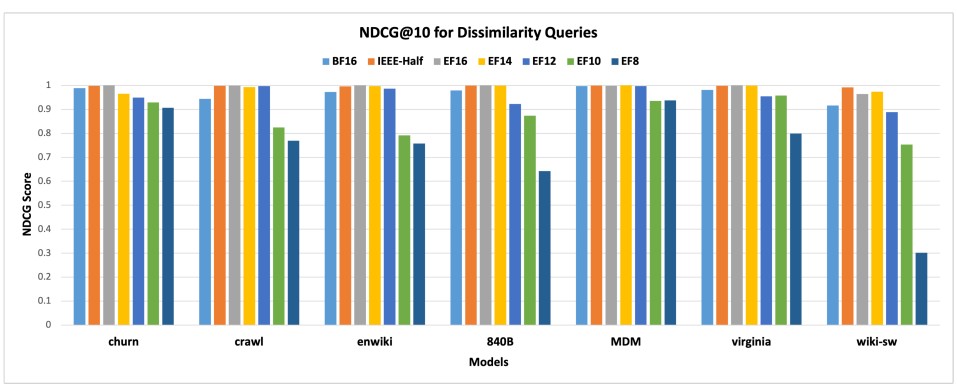

Figure 8: Evaluation of dissimilarity query accuracy using NDCG score across different floating point format. Higher score (closer to 1.0) is better.

## 6   Related Work

Model quantization is widely used to compress *pre-trained* models for the inference phase (Gupta and Agrawal (2020)). Quantization covers two broad approaches: the first represents a full-precision (e.g., 32-bit) floating point weight value using reduced (e.g., BF16 or FP16) or mixed precision floats, and the second converts full-precision floating point values into integer values with fewer bits (e.g., INT8, INT4, and INT1 (Migacz (2017); Wu et al. (2020); TensorFlow Documentation (2020); Jacob et al. (2017))). In conjunction with the model compression work, there has been significant work in devising reduced-precision floating point formats tuned for broader machine learning and HPC applications (Sapunov (2020); Abdelfattah et al. (2020)). Unlike the inference-focused model compression work, reduced-precision floating points are designed to work for both model training and inference phases. The most common reduced-precision floating point formats use 16 bits. Current 16-bit implementations include IEEE 754 half-precision (FP16); Brain Floating Point, BFloat16 (Wang and Kumar (2019); Kalamkar et al. (2019)); and Deep Learning Float (DLFloat) (Agrawal et al. (2019)), with 1 sign bit, 6 exponent bits, and 7 fraction bits. TensorFloat-32 (TF32) from Nvidia is a 19-bit format that combines 8 exponent bits from BFLOAT16 and 10 exponent bits from IEEE FP16 (Kharya (2020)). Hybrid Block Floating Point (HBFP) (Drumond et al. (2018)), Intel Nervana's Flexpoint (Koster et al. (2017)), and Microsoft MSFP (Rouhani et al. (2020)) formats combine the advantages of fixed point and floating point formats by splitting up the significand and the exponent part which is shared across multiple numeric values. Recent research proposals have described training of key deep learning models using even reduced precision floating point values (8- and 4-bit floats) (Sun et al. (2020); Wang et al. (2018); Cambier et al. (2020); Mellempudi et al. (2019)). Recently proposed AdaptiveFloat (Tambe et al. (2020)) is an inference-targeted floating-point format which maximizes its dynamic range at a network layer granularity by dynamically shifting its exponent range via modifications to the exponent bias and by optimally clipping (quantizing) its representable datapoints. Our proposed EFloat design practically achieves the same result without altering the exponent range and quantizing full-precision values.

## 7   Conclusion

We introduced EFloat, a novel entropy-coded variable length floating point format for deep learning applications. This format can be used for compressing a trained deep learning model, as well as for enabling more accurate model representations using reduced-precision floating point formats. While our intended use cases were initially for the database embedding (db2Vec) workloads, we demonstrate that the proposed format works effectively for other vector embedding models, and can be used for a much broader class of NLP models including transformer-based models. Broadly, EFloat may be used in deep learning applications where tradeoffs need to be made between range, precision, memory capacity and bandwidth savings. As a future work, we plan to explore the Benford distribution pattern (Benford (1938); Newcomb (1881)) exhibited by significands of vector embedding models (Appendix A in the supplementary document) and investigate its application in rounding EFloat values. A follow-up study on 8-bit floats and integers is being considered as well.

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
