# EFloat: Entropy-coded Floating Point Format for Deep Learning
# Supplementary Material

## A   Appendix

Figures 1 and 2 present patterns observed in the significands of FP32 values of various pre-trained vector embedding models used in the evaluation of EFloat format. Figure 1 plots the frequency distribution of the most significant significand digit in the decimal representation of the FP32 values. Occurrences of these digits follow a logarithmically decreasing distribution, called Benford's law (Newcomb (1881); Benford (1938)). According to Benford's Law, the number 1 occurs as the most significant digit about 30.1% of time, number 2 17.6% of time, and so on. The logarithmic reduction leads to the number 9 being at the most significant location around 4.6% of time.

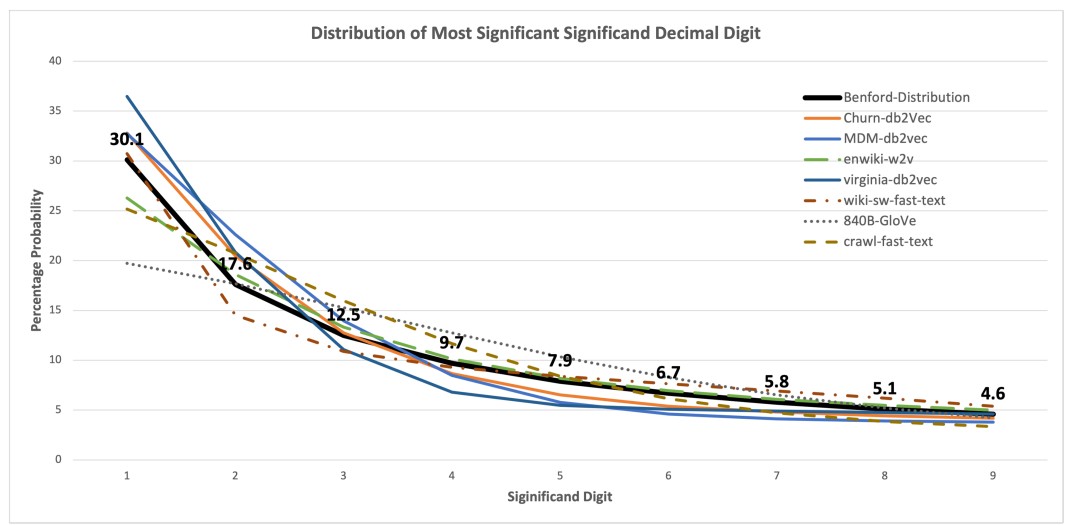

Figure 1: The most significant decimal digit in the significand of FP32 values in various vector embedding models follows the Benford distribution.

Figure 2 presents a bit-specific view of the same datasets. In this Figure, we report the agregate distribution of a particular bit location having the value 1 for 23 significand bits computed over all FP32 entries of the selected vector embedding models. As we observe, the distribution exhibits a knee at location 7; the first 7 bits show an increase in the probability of the bit value being 1 from 41.51% to 49.88%; the remaining bits exhibit probabilities around 50%. In other words, it is more likely to see bit values being 0 in the most significant bit locations of FP32 values in the pre-trained vector embedding models.

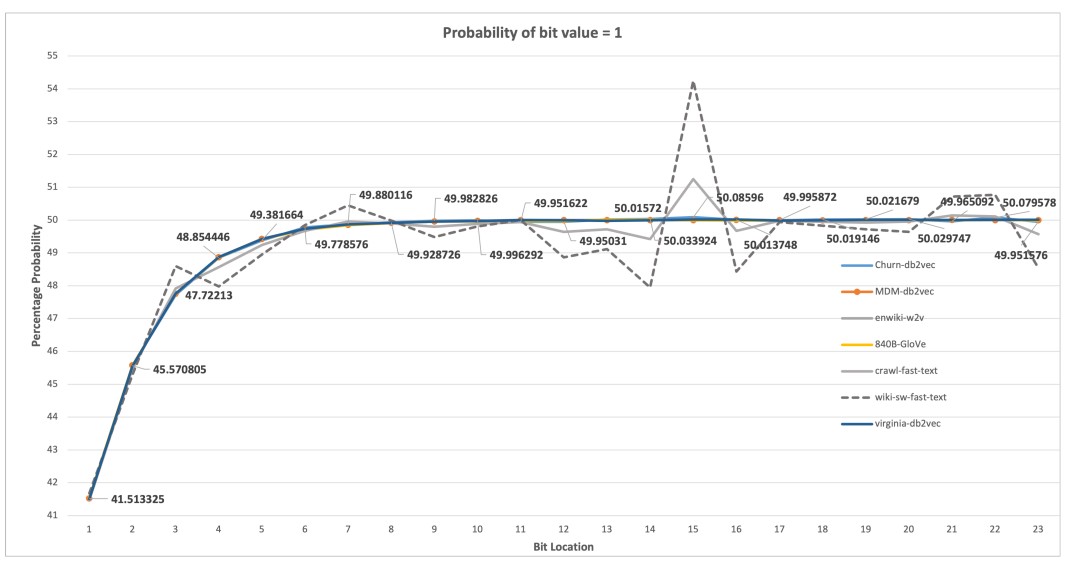

Figure 2: Distribution of a bit value being 1 at different bit locations in the significand of FP32 values in various vector embedding models.