# OpenReview forum: "EFloat: Entropy-coded Floating Point Format for Deep Learning"
_NeurIPS.cc/2021/Conference — NeurIPS 2021 Submitted_

### Official Review · Reviewer_f6Ju · 2021-07-16

**Rating:** 7
**Confidence:** 5

**Summary:**

The work proposes a new floating-point data type with entropy encoded exponent to store large embedding models. These data types are studied under several bit widths and compared with 16-bit brain float and IEEE 16-bit float data types and show favorable results.

**Limitations And Societal Impact:**

These are not discussed by the authors. I think the work has no negative societal impact, but a discussion of limitations would have been helpful.

**Main Review:**

Originality: Entropy encoding of the exponent is a well-studied method to compress floating-point data, but, to my knowledge, has not seen application in deep neural networks which usually have different data distributions than other domains of computer science and also have systematically shifting distributions over the course of training. This work provides some much-needed data on the effectiveness of exponent compression and how the underlying exponents shift during training. I see this work as an important contribution that will inform many other eep-learning-focused floating-point data types that are yet to come.

Quality: The manuscript has a high quality. It is easy to read and contains extensive explanations of core concepts which will make it easy to read for anyone outside of this sub-field. The quality of experiments is satisfactory, but the quality could be improved substantially by including other data types beyond the standard brain-float and IEEE data types. While other specific data types are a pain to implement, I would have wished to see 14 and 12-bit versions of IEEE and brain float data types in the comparison. This would have allowed one to make out trends in how these data types diverge in usefulness as the available bits are reduced.

What I am missing is a more careful analysis of approximation errors. Instead of RMSE ratio, I would have liked to see more standard absolute errors and relative/approximation errors. Just reporting one form of relative errors might be deceiving since values with very small magnitude usually have high relative errors but small absolute errors. Additionally, a break down of absolute errors per exponent would yield rich information of behavior across EFloats and other data types. Insights from such information would greatly benefit future work in this area.

Additionally, the generality of EFloats would be much clearer if it would be applied to different networks and tasks. Although the comparison with other data types that work seamlessly for training are favorable, from my experience new data types can include certain kinks which can introduce new problems during training. Such problems might not be apparent for the task explored in this paper since these networks are shallow compared to other mainstream neural networks.

Clarity: As mentioned before, the manuscript is easy to read and the details are perfectly clear. The work should be easily readable for researchers in different sub-areas.

Significance: I believe this work will inform any future work on floating-point data types for machine learning. As such, it is a significant contribution. However, compared to other work at NeurIPS, I can see that this work currently has a rather narrow focus and it is unclear if it would generalize to effective training of other networks, such as transformers or convolutional networks.

**Time Spent Reviewing:**

1

---

### Official Review · Reviewer_7rZw · 2021-07-16

**Rating:** 5
**Confidence:** 5

**Summary:**

The paper proposes a new tapered precision numerical format (EFloat) where the exponent of floating value is represented by Variable Length Huffman coding (The maximum of bit-width is limited). Since the Huffman coding was designed based on the probability of exponent occurrence, the most frequent exponent values are presented by fewer bits, which free some bits to be allocated for precision. The vector embedding benchmarks are used to evaluate the efficacy of the new numerical format compared to other numerical formats such as BFLOAT16 and FP16. Based on the metric represented in the paper, such as RMSE and NDCG, the EFloat is better than BFLOAT16 and comparable with FP16.

**Ethical Concerns:**

There are no ethical issues with this paper

**Limitations And Societal Impact:**

The authors did not address the limitations and potential negative societal impact of their work. The suggestion is presented in the Weakness section.

**Main Review:**

Strength:

1- Using Huffman coding for the exponent is an interesting and novel idea.

2- Benford’s low is interesting and is used in approximate computing, and I would suggest the author add this analysis instead of considering it as future work.

Weakness and suggestion:

1- Comparision with other numerical formats such as DFLOAT16, Posit [1], and Adaptive Posit [2] (Regime size is limited) is not discussed in the paper. The author needs to compare the new approach with these numerical formats experimentally.

2- NDGC and RMSE are not comprehensive metrics for comparing the new numerical format with others. One way of comparing the numerical format is decimal accuracy [1]. Furthermore, the author can also report log loss [3].

3- The trade-off between accuracy and hardware complexity is not discussed in this paper compared to other numerical formats.  The LUT-based conversion from EFLOAT to FP32 is explained in the paper; however, the complexity of this conversion does not quantify ( Experimentally or Theoretically )

4- It is not obvious why the FP16 showed better performance in terms of RMSE and NDGC metrics than EFloat (in some cases), and this limitation of EFloat is not discussed in the paper.

5- How the author ignoring the overflow in significant if all the bit is 1. Based on the explanation in the paper, if we increase the least significant bit, then we have an overflow. This part is unclear on the paper, and it would be suggested to be explained better. Moreover, why did the author not use round-to-nearest-even?

6- The collection of histograms is a complicated task, and as mentioned in the paper, the statistic of histograms needs to be considered. However, the author did not discuss this limitation in detail and the solution to overcome it.

7- The author suggests the log n +2 for the length of the exponent. However, this might be true for this application, but in general, the length depends on the application. What is the solution of the author for this problem? Is it needed to store the size of the exponent in a register?

8- Suggestion: Length-Limited Huffman algorithm is the suboptimal and theoretical analysis needed to be performed to understand the effect of this sub-optimal design on vector embedding performance. This theoretical analysis, such as theoretical upper-bound on convergence rate.

References:

1- Gustafson, John L., and Isaac T. Yonemoto. "Beating floating point at its own game: Posit arithmetic." Supercomputing Frontiers and Innovations 4.2 (2017): 71-86.

2- Langroudi, Hamed F., et al. "Adaptive Posit: Parameter aware numerical format for deep learning inference on the edge." Proceedings of the IEEE/CVF Conference on Computer Vision and Pattern Recognition Workshops. 2020.

3- Zhang, Jian, Jiyan Yang, and Hector Yuen. "Training with low-precision embedding tables." Systems for Machine Learning Workshop at NeurIPS. Vol. 2018. 2018.


Post Rebuttal=====
I appreciate the authors' responses to my comments. After carefully reading your responses and other reviewers’ comments, I have decided to increase my score to 5. However, still, I believe the experiment section lacks standard deep learning evaluation metrics as also mentioned by other reviewers. One suggestion is to use a hybrid EFloat and float16 for transformer or other modern NLP applications and report the standard NLP metric. Besides this, I still recommended comparing with DFloat16 and Posit (Adaptive Posit). I believe in a research study we try to compare the ideas that conceptually are the same but the approaches are different (It does not matter the approach is commercialized or which approach shows better performance). The Efloat and Posit (Adaptive Posit) conceptually are the same and both try to represent bell-shaped curve efficiently with the variable encoding of the exponent.


**Time Spent Reviewing:**

48

---

### Official Review · Reviewer_AEbL · 2021-07-17

**Rating:** 4
**Confidence:** 4

**Summary:**

This paper introduces EFloat, a domain-specific entropy-encoding numerical format targeted for NLP applications. EFloat takes advantage of clustered values in parameter distributions to allow Huffman coding to compress the average bitwidth. The authors propose this method as a means of compressing parameters to save memory and IO bandwidth in deep learning models. They further introduce the vector embedding model, describe the algorithms for encoding and decoding EFloat, and evaluate EFloat against IEEE FP16 and BFloat.

**Ethical Concerns:**

No extra ethical concerns except those found typically with machine learning.

**Limitations And Societal Impact:**

No extra negative societal impacts compared to those found typically with machine learning.

**Main Review:**

Overall, any method that decodes parameters before inference should be accompanied with a strong explanation of its hardware efficiency or a careful discussion of its limitations and intended domain. The current paper does not sufficiently motivate EFloat over standard table look-up quantization methods. Furthermore, the experimental section lacks standard deep learning evaluation metrics, and the RMSE values and NDCG scores given are not impressive enough in my opinion to deal with the extra complexity of EFloat.

Strengths:
1. The design of the encoding and decoding methods using code tables seems well-designed with efficiency in mind.
2. Figure 2 is a strong motivation for compressing the exponent bits.
3. This seems like a perfect application for length-limited Huffman coding.

Weaknesses:
1. "Deep Learning" title seems to overstate the domain discussed in the paper, which largely focused on vector embedding models.
2. Section 4. clearly states the encoding and decoding procedures and the small size of the code tables, but it would be helpful to have a better understanding of how this additional conversion affects the target hardware or applications. The paper focused primarily on the IO and memory bandwidth savings, yet the decoding and encoding task must have some significant performance tradeoffs even on specialized hardware backends.
3. The concept of entropy-encoded bitwidth seems to be general enough to map to any application once the data distribution is known. Since the relevant topics have been around for a half a century, this implies to me that there are some significant considerations that must be tackled to make such a method practical in hardware.
4. There should be some evaluation on standard accuracy metrics. Since this paper focuses its discussion on NLP models, perhaps there could be some comparisons using modern NLP models and datasets. The NDCC goes beyond the average bitwidth and RMSE scores but still falls short for me in evaluating the whole system.
5. The results in Figures 7 and 8 seem not strong enough to warrant the extra encode / decode complexity and uncertainty in the lossy compression.
6. There have been previous attempts to use Huffman coding on model parameters, e.g. Deep Compression, that should probably be included in the related works.
7. This is almost a hybrid method between lookup-based quantization for the exponent and floating point for the significant. There should be some at least high level comparisons to lookup-based quantization methods.

Questions
1. Were other compression schemes with the prefix property considered?
2. Are other more accurate compression methods for the exponent possible given a representative dataset? This method seems to be designed for pre-trained models.
3. Why the particular focus on the vector embedding models?
4. In my experience, most deep learning models can be quantized to 8 bits today with degradation in accuracy. Why the particular focus around 16 bits?


Minor Issues
1. L31 "it's" -> "its"

**Time Spent Reviewing:**

3-4

---

### Decision · Program_Chairs · 2021-09-27

**Decision:**

Reject

**Comment:**

This is a promising paper that studies an interesting idea. The reviewers' scores were mixed, but on balance the concerns about the experimental evaluation caused most reviewers to lean towards rejection. The paper would be substantially improved by more standard end-to-end performance comparison. For example, a strong empirical result would show that, for some hardware and some standard network, EFloat can run inference for a pretrained model faster than standard compression approaches (due to the reduced memory load). As written, we don't know when the trade-off made by EFloat will be useful, both because of the lack of standard end-to-end accuracy benchmarks and the incomplete evaluation of systems/hardware metrics.